# Weekday-Weekend Sedentary Behavior and Recreational Screen Time Patterns in Families with Preschoolers, Schoolchildren, and Adolescents: Cross-Sectional Three Cohort Study

**DOI:** 10.3390/ijerph18094532

**Published:** 2021-04-24

**Authors:** Dagmar Sigmundová, Erik Sigmund

**Affiliations:** Faculty of Physical Culture, Palacký University Olomouc, 77147 Olomouc, Czech Republic; dagmar.sigmundova@upol.cz

**Keywords:** screen time, daughter, son, mother, father, weekdays, weekends, preschoolers

## Abstract

Background: Excessive recreational screen time (RST) has been associated with negative health consequences already being apparent in preschoolers. Therefore, the aim of this study was to reveal parent-child sedentary behavior, and RST patterns and associations with respect to the gender, age category of children, and days of the week. Methods: Our cross-sectional survey included 1175 parent-child dyads with proxy-reported RST data collected during a regular school week during the spring and fall between 2013 and 2019. The parent-child RST (age and RST) relationship was quantified using Pearson’s (r_P_) correlation coefficient. Results: Weekends were characterized by longer RST for all family members (daughters/sons: +34/+33 min/day, mothers/fathers: +43/+14 min/day) and closer parent-child RST associations than on weekdays. The increasing age of children was positively associated with an increase in RST on weekdays (+6.4/+7.2 min per year of age of the daughter/son) and weekends (+5.8/+7.5 min per year of age of the daughter/son). Conclusions: Weekends provide a suitable target for implementation of programs aimed at reducing excessive RST involving not only children, but preferably parent-child dyads.

## 1. Introduction

Sedentary behavior and screen time, as one of the components of repetitive 24-h behavior, such as physical activity, eating, and sleep, are closely linked to the health of children and young people [1,2,3,4]. A longer duration of screen time (including television viewing, video games, and computer use) is significantly associated with higher cardiometabolic risks, lower fitness, self-esteem, and adiposity in preschoolers and schoolchildren [1,2,3] with obvious differences between girls and boys [4,5]. Although studies in preschoolers are less frequent than in schoolchildren and adolescents, it is documented that excessive screen time is associated with higher risk for overweight and obesity [6], and decreased scores of psychosocial health and cognitive development [2]. An updated systematic review of the associations between sedentary behavior and health indicators in preschoolers [3] confirms the results of a previous systematic review [2] and continues to support the minimization of screen time for disease prevention. Moreover, this review encourages the discovery of moderators of health promotion, with respect to gender, and highlights the potential cognitive benefits of interactive non-screen time behaviors, such as reading and storytelling [3]. It should be noted that only studies from Northern and Western European countries were represented in these systematic reviews [2,3]. Central and Eastern European countries were not included at all, although the prevalence of overweight and obesity, as well as the screen time of preschoolers from Central and Eastern Europe, is higher than among preschoolers from Northern and Eastern European countries [7,8].

Parents as “gatekeepers” of children’s health-related behavior [9] can best support their children in compliance with sleep rules and restricting excessive RST, while support in compliance with physical activity guidelines is weaker [10]. Nevertheless, the direct involvement of parents in the process of treating childhood obesity in accordance with Family Development Theory leads to an effective reduction of the excessive Body Mass Index (BMI) of schoolchildren and adolescents [11,12,13]. Subsequently, only compliance with the RST recommendations of the 24-h movement guidelines was significantly associated with reduced odds of a high BMI z-score, excess fat mass%, and visceral adipose tissue [14]. Although the above findings concerned schoolchildren and adolescents [11,12,13,14], it can be reasonably assumed that the amount of daily RST as well as the involvement of parents in controlling sedentary behavior of their offspring also play an important role in the prevalence of overweight in preschool children.

Similar to schoolchildren and adolescents, preschoolers spend more than 50% of their objective monitored waking hours being sedentary regardless of the day of the week; there is an observable gender difference though, as girls are more sedentary than boys [15]. While many correlates of sedentary behavior have been highlighted in schoolchildren and adolescents [1,16], recent systematic review was unable to identify any consistent correlates of sedentary time in preschoolers other than positive associations with parental sedentary behavior [17]. Repeatedly reported correlates associated with excessive screen time in schoolchildren and adolescents include the number of accessible screen-time devices, the presence of a television in the bedroom, fewer family rules about television viewing, infrequent family meals, less walkable neighborhoods, fewer appealing outdoor areas, and concerns about neighborhood safety [16]. However, objective instrumental monitoring of sedentary behavior in preschool children is not yet widespread, as in schoolchildren and adolescents, or does not cover weekdays and weekend days with a limited differentiation of sitting and reclining positions [18]. From the existing parent-reported screen time studies, positive relationships between parents’ and their preschooler children’s screen time, with differences by gender and days of the week, are evident [19,20,21,22]. However, these studies [19,20,21,22] used only a non-continuous categorized screen time score primarily for the analysis of compliance with screen time-related guidelines: max. one hour per day for preschoolers and max. two hours per day for older children and parents [23,24].

In the countries of Central Europe, including the Czech Republic, there are incomplete findings about sedentary behavior and especially the RST of families with preschool children, regarding the days of the week or the gender of family members. Therefore, the aims of this study are to: (i) reveal parent-child sedentary behavior and weekday-weekend RST patterns in family members with preschoolers in comparison with families with schoolchildren and adolescents; (ii) find out the associations between RST/BMI/age of family members; (iii) quantify the change in RST with increasing age of children. In this study, we will test the hypotheses: (i) whether there is a difference in the daily sedentary time ((ii) in the daily RST) in daughters and sons (mothers and fathers) between weekdays and weekends in families separated by age categories of offspring; (iii) whether the parent-child association in daily RST varies with respect to the days of the week and gender; (iv) whether there is a significant association between daily RST and BMI ((v) daily RST and age) of children with respect to the days of the week. The sedentary behavior of children is influenced by many more variables besides parental behavior on weekdays (e.g., kindergarten program, school regimen, or participation in leisure-time organizations, clubs, sports), while at weekends their parents apparently might have more chances to stimulate health-enhancing behaviors in their offspring. Therefore, we expect closer association between the RST of parents and their children on weekends than on weekdays [21,25].

## 2. Materials and Methods

### 2.1. Design of the Study and Ethics

This is a cross-sectional study in which, using the same methodology, parent-child indicators were cross-sectionally monitored in three groups of families (the first group—families with preschool children, the second group—families with children aged 7–11.9 years, and the third group—families with adolescent children aged 12–16) from the Czech Republic. The study design, content and format of the family logbook, feedback for participants, and method used for all the measurements were approved by the Ethics Committee of the Faculty of Physical Culture review board for families with school-aged children (reference number 50/2012) on 12 December 2012, and for families with pre-school children (reference number 57/2014) on 21 December 2014. The ethical principles of the 1964 Helsinki Declaration and its later amendments were adhered to throughout the research. Written informed consent was obtained from all participants and their parents (guardians) prior to the start of the data collection.

### 2.2. Selection of the Participants and Procedure

Invitation letters were sent to 3540 families from the Czech Republic, of which 65.3% agreed to take part in the research (written informed consent received). Families were selected through two-stage random sampling. In the first stage, nine out of 14 administrative regions, three of each in the highest, middle, and lowest terciles for gross domestic product in the Czech Republic, were randomly recruited. In the second stage, three public kindergartens located in rural areas and seven in urban locations and 15 public “basic schools” located in rural areas and 36 public “basic schools” in urban locations were randomly selected. Random selection of administrative regions and subsequently kindergartens and basic schools was performed using a random number generator from tables in computer memory. The “basic schools” in the Czech Republic provide compulsory education from grades 1 to 9. The participants were largely white Caucasian (>98%), which is representative of the ethnic demographics of the Czech Republic [26].

All the participants who confirmed their participation in the research in writing were acquainted in detail with the course of the monitoring of sedentary behavior and the method used for determining the body height and weight during an information meeting with the researchers in schools. Each family received an envelope with a family logbook for recording the anthropometric data (gender, calendar age, body height, and weight) and sedentary behavior of all family members. Data were collected in three stages during the spring and autumn months between 2013 and 2019 under comparable daily climate conditions. In the first stage, sedentary behavior was observed in families with children aged 7–11.9 years (April–May and September–October, 2013). In the second stage, the research was conducted in families with preschoolers (September–October, 2014 and April–May, 2015). Lastly, the research was carried out in families with children aged 12–16 years (September–October, 2018 and 2019 and April–May, 2019). Study flowchart of participants can be found at Appendix A.

### 2.3. Measures

Together with the researchers, the parents filled in the age and gender identifiers of the family members in the first part of the family logbook and were introduced to the second part of the family logbook for recording sedentary behavior. The second part of the logbook, concerning sedentary behavior, consisted of seven items (Table 1). The RST represents the sum of two types of recorded sedentary behavior, sitting/lying watching television (videos) and sitting/lying in front of a computer screen (notebook, tablet, smartphone) not for school/work purposes. The duration and type of sedentary behavior were recorded in the second part of the family logbook by the parents, together with their children, each evening. The accuracy of the recording of the duration of each type of sedentary behavior was fixed at 10 min. The parents’ proxy reporting of the time their five-to-six-year-old children spent watching television each day exhibits an acceptable 7–4-day test-retest reliability (ICC = 0.78, *p* < 0.001) for capturing sedentary behavior on regular weekdays and weekend days [27]. The full list of anthropometric and sedentary behavior-related items of family members recorded by parents is depicted in Table 1.

Anthropometric measurements were conducted in the participants’ homes. The parents were instructed in detail by the researchers how to measure their own body height and weight, as well as the height and weight of their children, through an enclosed illustrated instruction leaflet for home measurement [28]. The instruction leaflet for home measurement of the body height and weight, the correct upright posture against a wall (barefoot), and the correct reading of the resulting body height were depicted. The body weight measurements were illustrated barefoot and with the participants only in their underwear. The parents recorded the body height and weight values to the nearest 0.5 cm and 0.5 kg in the first part of the family logbook. Parental measurements of children’s body height and weight at home demonstrate almost perfect agreement with direct measurements of body height (using a portable rigid stadiometer) and body weight (Tanita weight scale) [29] and the BMI derived from home measurement of body height and weight shows good diagnostic ability for identifying underweight and overweight/obesity categories in children [29,30].

### 2.4. Data Management and Statistical Analysis

The criteria for inclusion of the data in the final set were as follows: (i) complete anthropometric data (year and month of birth, gender, body height, and weight); (ii) attendance in kindergarten or basic school according to schedule (children), paid employment (parents other than maternity leave) on at least four school/working days a week; (iii) recording of the structure of sedentary behavior (type, duration) on at least four school/working days and one weekend day; (iv) at least one parent-child pair (mother-child, father-child) per family. Inclusion criteria for the minimum number of days per week for a valid capture of sedentary behavior were determined according to the recommendations of thematically related studies in children [31,32] and adults [33,34].

After checking for extreme and erroneous values for the anthropometric indicators, the BMI values for each family member were calculated separately as the body weight (kg) divided by body height (m) squared. Similarly, the daily sedentary time/RST values were checked for extreme and erroneous data, and the average daily sedentary time/RST was calculated separately for weekdays and weekends. To maintain the comparability of family-related sedentary behavior data with previous studies focused on pedometer-based family physical activity [21,28,35], the same procedure for controlling and calculating anthropometric indicators and sedentary time/RST variables was retained. If daily sedentary time /RST was recorded during four weekdays, data for the one missing weekday that were based on the participant’s personal mean scores were added. Those participants whose daily sedentary time /RST data were missing for more than one day were excluded from the analysis [31,32,33,34]. The average daily sedentary time/RST was calculated separately for weekdays and for weekends as the sum of the individual daily sedentary time/RST divided by the appropriate number of days.

The normality of the distribution of daily sedentary time/RST variables was tested using the Shapiro–Wilk and Kolmogorov–Smirnov tests. Neither the Shapiro–Wilk test nor the Kolmogorov–Smirnov test confirmed the normal distribution of daily sedentary time/RST variables. As a result of the non-normal distribution of daily sedentary time and daily RST, the Wilcoxon test was used to compare weekday-weekend daily sedentary time/RST in each of the participants of families with preschoolers, 7–11.9 year old children, and 12–16 year old adolescents. Descriptive characteristics for the daily sedentary time/RST are presented in the form of means and a 95% confidence interval. Bivariate Pearson correlations (r_P_) were conducted that examined the association between the parents’ and children’s RST/BMI/age separately according to the gender of the participants and days of the week. To quantify the change in daily RST with increasing age of children, linear regression analyses were performed separately for weekdays and weekends.

The Statistical Package for the Social Sciences (SPSS) for Windows v.22 software (IBM Corp. Released 2013. Armonk, NY, USA) was used for all data management and all statistical analyses. The alpha level of significance was set at the minimum value of 0.05 for all the statistical analyses.

## 3. Results

Research data were received from 1,899 families, and 89 distant relatives, teachers, grandmothers, etc., who were excluded from the study. Of the total number of families, 724 families were excluded for non-compliance with any of the following inclusion criteria: (i) the impossibility of linking the parent-child sedentary behavior/RST record (*n* = 88), (ii) children <4 years old or ≥16 years old (*n* = 151), (iii) missing data about gender (*n* = 64), (iv) missing data about body height or weight (*n* = 169), (v) insufficient number of days with a sedentary behavior/RST record that covered less than four working days and one weekend day (*n* = 252). The final dataset contained 1,175 families (179 families with preschoolers, 665 families with children aged 7–11.9 years, and 331 families with adolescents aged 12–16 years) with complete and correct anthropometric and sedentary/RST data. The summary sample anthropometric characteristics of the final set of participants are presented by means and standard deviations in Table 2.

### 3.1. Weekday-Weekend Patterns of Sedentary Behavior and RST of Family Members

#### 3.1.1. Daughters and Sons

Different weekday-weekend day patterns of overall sedentary behavior were detected in offspring in different age categories. While for preschoolers, both daughters and sons, there is no obvious difference in the duration of daily sedentary time between school and weekend days, for schoolchildren, both girls and boys, significantly (*p* < 0.001) more sedentary time is evident on school days than on weekends (Figure 1). Moreover, the overall daily sedentary time of the 7–11.9 year old schoolchildren on school days exceeds the sedentary time on weekends by more than 75 min on average, and for the12–16 year old adolescents this difference is already more than 100 min (Figure 1).

However, in the case of daily RST, the weekday-weekend day patterns in all the age categories of children that were analyzed are very different from the weekday-weekend patterns of their sedentary time (Figure 1 and Figure 2). The weekend daily RST is significantly longer in all age categories of children (*p* < 0.001) than the daily RST on school days (daughters: on average by 32 to 38 min per day; sons: on average by 24 to 39 min per day). Moreover, for the comparison of the daily RST from the youngest to the oldest age category of the offspring, a typical upward trend is typical, i.e., the amount of all-day RST for preschoolers on weekends corresponds to the all-day RST for 7–11.9 year old schoolchildren on school days and their significantly higher (*p* < 0.001) amount of the RST on weekends corresponds to the amount of RST for adolescents of older school age on school days (Figure 2).

#### 3.1.2. Mothers and Fathers

For the parents of children in all the age categories that were analyzed, the total daily sedentary time on weekdays is significantly (*p* < 0.001) longer than the total daily sedentary time on weekends (mothers: on average by 84 to 126 min per day; fathers: on average by 94 to 141 min per day) (Figure 3). Regarding the daily RST of parents, as with children, a longer daily RST on weekends than on weekdays is also evident, but with some differences between mothers and fathers (Figure 4). Only in the mothers of preschool children was a significantly higher RST revealed on weekends than on weekdays. However, for both the mothers and fathers of older children and adolescents, an RST was evident that was significantly longer on weekends than on weekdays (mothers: on average by 45 to 55 min per day; fathers: on average by 14 to 23 min per day) (Figure 4).

### 3.2. Associations between Daily Parent-Child RST, and Children RST, BMI, and the Calendar Age

The analysis of the associations between the daily RST of parents and their offspring pointed to closer associations between mothers and children of both genders than fathers and their offspring, and closer associations between the daily RST of parents and their offspring on weekends than on school days/workdays (Figure 5). The closest associations between daily RST in mothers and their offspring were found in preschoolers and the least close mother-offspring associations in RST in families with adolescents. In mothers and their offspring, both daughters and sons, a significant (*p* < 0.01) association between the daily RST and the BMI, was confirmed (r_PMOTHER x DAUGHTER_ = 0.228, r_PMOTHER x SON_ = 0.181). In addition, a significant (*p* < 0.01) association between the daily RST and calendar age was also confirmed in the children (r_PDAUGHTER_ = 0.251, r_PSON_ = 0.295).

### 3.3. Associations between Children’s Daily RST and Calendar Age

The results of a more accurate linear logistic regression analysis of the association between RST and children’s age, which allows the quantification of the potential minute changes in daily RST with increasing age of children, are presented in Figure 6. Given the positive significant associations between daily RST and children’s age (Figure 5), a significant increase in daily RST is expected with increasing age of the children. Linear logistic regression analysis confirms a significant increase in daily RST with increasing age of daughters and sons. The course of the daily increase in RST is similar on school and weekend days—for daughters/sons on average by about six or seven minutes per calendar year. Thus, during childhood, between four and 16 years of calendar age, the daily RST potentially increases by more than 70 min per day for daughters and by approximately 90 min per day for sons (Figure 6).

## 4. Discussion

The key findings springing from the results are as follows: (a) completely different weekday vs. weekend patterns of sedentary time versus RST in all family participants; (b) the lowest daily time spent on RST in preschool children, with a clear increase in daily RST in older children aged 7–11.6 years and 12–16 year old adolescents on both school and weekend days; (c) significantly close parent-child RST associations on weekends, regardless of the gender of family members. Consistent monitoring and analysis of the sedentary behavior of offspring, and especially their ST, is important because of the 24-h movement guidelines (i.e., physical activity, sedentary behavior, and sleep); parents can best support the sleep guidelines and ST restriction rules [10]. In addition, in 14–18 year old Czech adolescents, it was revealed that compliance with only the ST recommendations of the 24-h movement guidelines was positively associated with reduced odds of a high BMI *z*-score (odds ratio [OR] = 0.38, 95% confidence interval [CI]: 0.17–0.89), excess fat mass% (OR = 0.34, 95% CI: 0.13–0.93), and visceral adipose tissue (OR = 0.27, 95% CI: 0.10–0.74) [14].

Previous studies have shown that preschool children have a pedometer-measured daily number of steps comparable to those of school-aged children and adolescents, and the closest association in parent-child daily steps among families with preschool and older children and adolescents [35]. However, daily sedentary time and RST have not yet been compared across the age spectrum of children in the Czech Republic regarding parental sedentary behavior [21,35]. Similar to the results from Australia, the United States, or Canada, the lowest levels of sedentary time and RST were found in preschool children compared to older children and adolescents, as well as there being a positive association between age and RST in children [16]. Consistently with previous studies [20,36], different weekday-weekend patterns of sedentary time and RST in older children and adolescents have been demonstrated, but no significant gender-related differences in sedentary time and RST in older children and adolescents were revealed. Following a comprehensive study of the sedentary behavior of adolescents [16], which reveals that adolescents are the most sedentary pediatric population and the one most involved in RST, we add that this finding also applies to the parents of adolescent offspring.

However, the RST of children and adolescents has undergone a significant change in the last 20 years; traditional television/video viewing has been replaced by computer- or video game-based screen time and social media-based screen time [37]. Following these changes in children’s RST patterns, many important family, home, and neighborhood environment correlates have been identified. The presence of a computer, television, or video game system in the bedroom is positively associated with children’s RST, as is the number of computers, televisions, or game consoles in the household [38,39,40]. Lower frequency of family meals and eating meals in front of the television are associated with longer ST in adolescents [41]. On the other hand, for example, more outdoor play, the application of parental television viewing rules, or living in neighborhoods with more walking infrastructure, services, and parks has been associated with shorter RST in children [39,42,43]. However, the limit of studies [38,43] is the non-inclusion of children’s sex in the analysis, and in study [42], estimates were similar for boys and girls, although some associations were no longer significant in the sex-stratified models. In mothers of preschool children, there is a close association between their television viewing and screen media use in their offspring, but in addition maternal distress or depression and less cognitive stimulation in the home environment is associated with longer screen time for preschoolers [44]. Although we did not focus on detecting correlates other than parental sedentary time correlates in this study, it is important to point out the above-mentioned correlates of RST, especially in families with preschool children, as the increase in sedentary time (or RST) between four and 16 years of age may be more than 100 (70) min for daughters and 120 (90) min for sons.

RST in children and adolescents is associated with adiposity [14,45,46,47] and metabolic syndrome [45,48] and this association often persist after adjustment for physical activity and diet [16,48]. Although the adiposity of participants in families was not evaluated, the significant positive relationship between BMI and RST in daughters, sons, and mothers supports the above findings. It is clear that every child needs to be involved in some RST per day, but for the primary prevention of obesity, it may be important to promote sedentary habits at short intervals and to limit prolonged time spent in front of the device screen/display [47,49].

A representative set of Czech families, the recording of sedentary behavior-related items as continuous variables, and the relatively strict criteria for inclusion in the cross-sectional study are the strengths of the study. However, it is necessary to list the limits (the potential effect of social desirability and the degree of conscientiousness of proxy-recording of anthropometric and ST-related variables), which arise from the non-instrumental way the sedentary behavior of the participants was recorded. Although the variables related to sedentary behavior and RST were clearly differentiated, possible multitasking was not observed in detail (i.e., simultaneous use of multiple screen devices or different activities on one screen at the same time) [50]. However, for all the participating families, the methodology used was applied uniformly, which allowed a cross-sectional comparison of sedentary behavior and screen time. The associations of parent-child sedentary behavior and RST patterns could have been influenced by other lifestyle factors, such as parents’ occupation and education or family eating habits, which were not monitored in the present study. Future studies should, therefore, account for these factors too.

The new space for detecting parent-child gender-separated patterns of behavior opens up 24-h monitoring of behavior based on the use of multifunctional devices [14,47], which enable the accurate capturing of sedentary behavior and movement activities with regard to speed or intermittent execution, localization, or joint implementation [16].

## 5. Conclusions

Completely different weekday vs. weekend patterns of sedentary time and daily RST in all family members, the specifics of sedentary behavior in families with preschoolers, and the closest parent-child RST associations on weekends should be respected and taken into account in creating and applying programs to reduce sedentary behavior. A significant increase in daily RST on weekends compared to school days for daughters and sons of all ages, which has replaced sedentary time on school days for schoolchildren, is a critical indicator of a sedentary lifestyle. Thus, weekends provide a suitable target for implementation of programs aimed at reducing excessive RST involving not only children, but preferably parent-child dyads.

## Figures and Tables

**Figure 1 ijerph-18-04532-f001:**
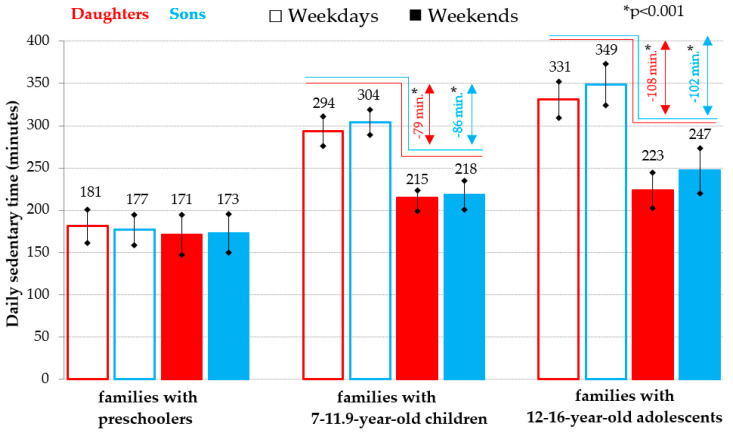
Weekday-weekend pattern of overall daily sedentary time (mean and 95% confidence interval) in daughters and sons in families divided according to the age of the offspring.

**Figure 2 ijerph-18-04532-f002:**
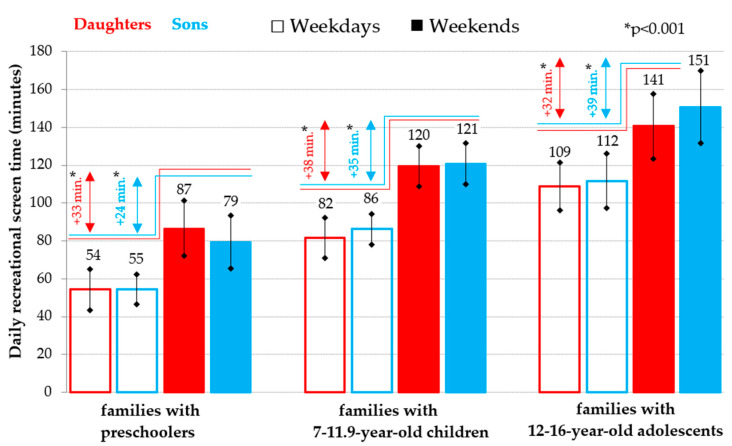
Weekday-weekend pattern of daily recreational screen time (mean and 95% confidence interval) in daughters and sons in families divided according to the age of the offspring.

**Figure 3 ijerph-18-04532-f003:**
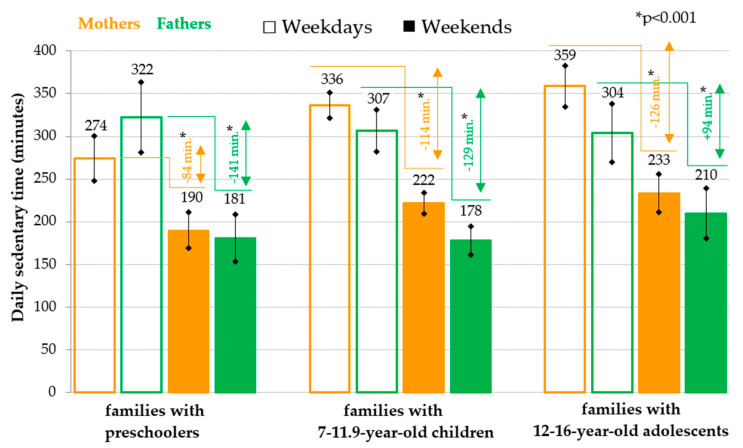
Weekday-weekend pattern of overall daily sedentary time (mean and 95% confidence interval) in parents divided according to the age of the offspring.

**Figure 4 ijerph-18-04532-f004:**
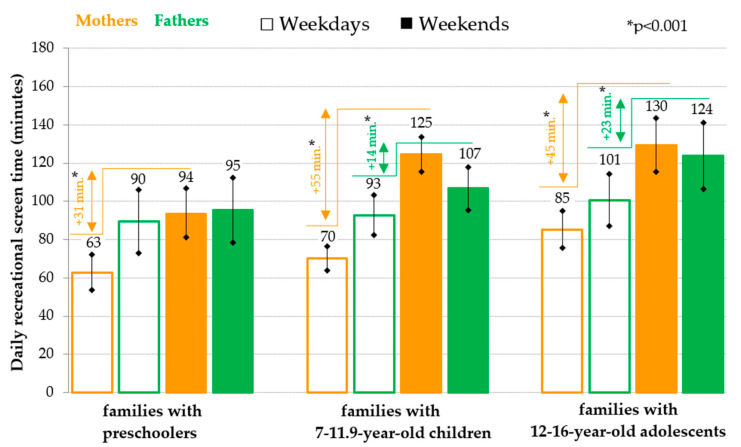
Weekday-weekend pattern of daily recreational screen time (mean and 95% confidence interval) in parents divided according to the age of the offspring.

**Figure 5 ijerph-18-04532-f005:**
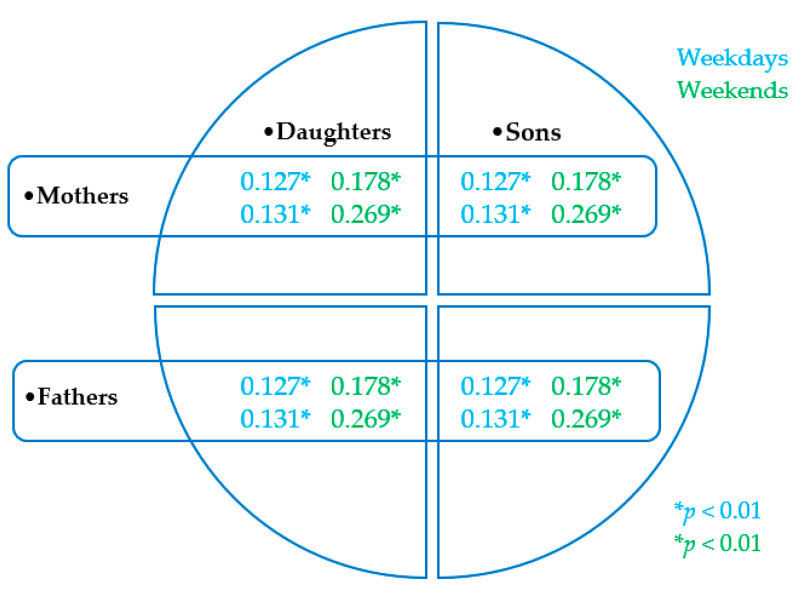
Parent-child associations (r_P_) in recreational screen time at weekdays and weekends.

**Figure 6 ijerph-18-04532-f006:**
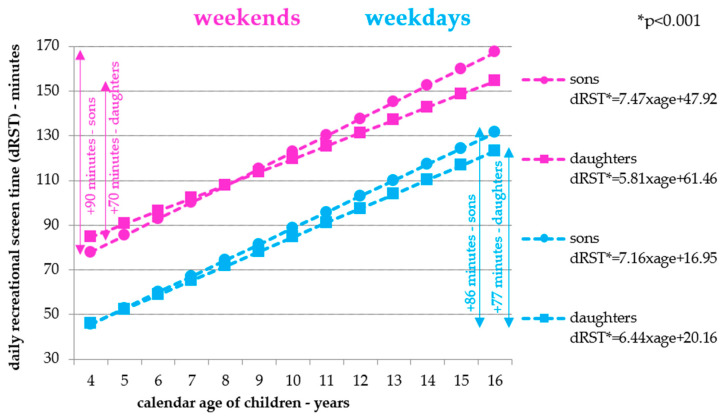
Associations between children’s daily recreational screen time and calendar age.

**Table 1 ijerph-18-04532-t001:** Parent reported measured anthropometric and sedentary behavior related items.

Item Type	Measurement	Units
Anthropometric		
Calendar age	Date of birth (children), age (parents)	years and months
Gender	Female, male	
Body height	home measurement—correct upright posture against a wall (barefoot)	nearest 0.5 cm
Body weight	home measurement—participants only in their underwear (barefoot)	nearest 0.5 kg
Sedentary-related		
1	sitting/lying watching television (videos) not for school/work purposes	minutes for the whole day
2	sitting/lying in front of a computer screen (notebook, tablet, smartphone) not for school/work purposes	minutes for the whole day
3	sitting/lying when studying, reading, and playing (non-computer games, a musical instrument, drawing, and painting)	minutes for the whole day
4	sitting in a park (shopping center, restaurant)	minutes for the whole day
5	sitting in a movie theatre (theatre, concert)	minutes for the whole day
6	sitting in a vehicle (car, bus, train, streetcar)	minutes for the whole day
7	sitting in school/kindergarten (paid employment for parents)	minutes for the whole day

**Table 2 ijerph-18-04532-t002:** Final set of participants aggregated according to the age category of children and separated by gender.

Age Category of Children	Age Number	Daughters (mean ± SD)	Sons (mean ± SD)	Mothers (mean ± SD)	Fathers (mean ± SD)
4–6.9 years	Age (years)	5.78 ± 0.77	5.80 ± 0.68	37.20 ± 3.99	40.02 ± 4.99
	Number	*N* = 98	*N* = 91	*N* = 171	*N* = 118
7–11.9 years	Age	9.47 ± 1.48	11.92 ± 1.43	39.25 ± 3.98	41.78 ± 5.18
	Number	*N* = 256	*N* = 255	*N* = 439	*N* = 272
12–16 years	Age	13.58 ± 1.06	13.45 ± 1.03	42.07 ± 4.72	44.49 ± 5.88
	Number	N = 154	*N* = 135	*N* = 224	*N* = 164

Legend: SD—standard deviation.

## Data Availability

The data are owned by Palacký University Olomouc and are not to be made freely publicly available during the project investigation but are available from the corresponding author E.S. upon reasonable request.

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
