# Peer review of "Weekday-Weekend Sedentary Behavior and Recreational Screen Time Patterns in Families with Preschoolers, Schoolchildren, and Adolescents: Cross-Sectional Three Cohort Study"

_ijerph, 2021, doi:10.3390/ijerph18094532_

Round 1
Reviewer 1 Report
The present study revealed the association of sedentary behavior between children and parents using a Chechia representative dataset. Comparisons of the associations between weekday and weekend, sedentary behavior and recreational screen time, daughters and sons, and mothers and fathers are fascinating, and the manuscript is beautiful. I have some minor comments as follows.
- In the abstract, the authors referred to only recreational screen time; however, in the title, they referred to sedentary behavior and recreational screen time. Please include sedentary behavior in the abstract.
- In the introduction, there were so many “however.” Please revise them.
- In table 1, there was an error in describing the number of sons of 4-6.9 years (N±91).
- For comparisons between mothers and fathers, whether they work or not might associate with children’s sedentary behavior. Did the authors collect the parents’ occupation?
Author Response
(Reviewer 1): The present study revealed the association of sedentary behavior between children and parents using a Chechia representative dataset. Comparisons of the associations between weekday and weekend, sedentary behavior and recreational screen time, daughters and sons, and mothers and fathers are fascinating, and the manuscript is beautiful. I have some minor comments as follows.
We are glad that the reviewer saw the potential of our work and appreciate the comments provided.
- In the abstract, the authors referred to only recreational screen time; however, in the title, they referred to sedentary behavior and recreational screen time. Please include sedentary behavior in the abstract.
Thank you very much for this comment. Consistent with similar comments by Reviewer 3 and 4, we have reformulated and clarified the text of the entire abstract.
Abstract: Background: Excessive recreational screen time (RST) has been associated with negative health consequences already being apparent in preschoolers. Therefore, the aim of this study is to reveal parent-child sedentary behavior and RST patterns and associations with respect to the gender and age category of children and days of the week. Methods: Our cross-sectional three-cohort survey included 989 parent-child dyads with proxy-reported RST data collected during a regular school week during the spring and fall between 2013 and 2019. The parent-child RST (age and RST) relationship was quantified using Pearson's rP correlation coefficient. Results: Weekends are characterized by longer RST for all family members (daughters/sons: +34/+33 minutes/day, mothers/fathers: +43/+14 minutes/day) and closer parent-child RST associations than on weekdays. The increasing age of children is positively associated with an increase in RST on weekdays (+6.4/+7.2 minutes per year of age of the daughter/son) and weekends (+5.8/+7.5 minutes per year of age of the daughter/son). Conclusions: Weekends provide a suitable target for implementation of programs aimed at reducing excessive RST involving preferably all family members, which holds true both for families with preschoolers and for families with schoolchildren.
- In the introduction, there were so many “however.” Please revise them.
Many thanks for this comment. In line with similar comment by Reviewer 4, we have reformulated and clarified the full text of the Introduction (including the less frequent use of the term ‘however’).
- In table 1, there was an error in describing the number of sons of 4-6.9 years (N±91).
Thank you for this noticing that. We have replaced the ‘±’ symbol by ‘=’.
- For comparisons between mothers and fathers, whether they work or not might associate with children’s sedentary behavior. Did the authors collect the parents’ occupation?
Thank you for pointing this issue out. To reflect on the comment, we did not monitor the parents’ occupation. Therefore, we have added the following text to the study limitations.
The associations of parent-child sedentary behavior and RST patterns could have been influenced by other lifestyle factors such as, parents’ occupation and education or family eating habits, which were not monitored in the present study. Future studies should therefore account for these factors too.

Reviewer 2 Report
The only item I feel you need to address is on lines 72, 87, & 91 when ‘Czechia’ is mentioned. From reviewing the manuscript you (the authors) are from the Czech Republic. Thus, this term may be a common term to refer to your country. I suggest you change it to the Czech Republic. The readers may not understand.
Author Response
(Reviewer 2): The only item I feel you need to address is on lines 72, 87, & 91 when ‘Czechia’ is mentioned. From reviewing the manuscript, you (the authors) are from the Czech Republic. Thus, this term may be a common term to refer to your country. I suggest you change it to the Czech Republic. The readers may not understand.
Thank you for seeing the potential of our work. We acknowledge that the term ‘Czechia’ is not that widespread, even though it is an official geographical name registered by the UN. To avoid any confusion, we have changed Czechia to Czech Republic throughout the text of the manuscript.
Reviewer 3 Report
This is an interesting study comparing sitting behavior and recreational screen time for Czech parents and children. Since it provides valuable knowledge, I believe that the possibility of publication will increase by making the following modifications.
- The conclusions of the abstract should be significantly revised. The reason is that it is a cross-sectional study, and since it is not based on causality, it is highly likely that the reader will misunderstand it.
- The terms randomly are used on lines 88 and 90, but details should be given to what population they were randomly sampled from and how they were randomly sampled. Being a representative of the Czech people should show the rationale.
- Insufficient data on sitting behavior in line 99 states less than 4 days on weekdays and less than 1 day on weekends, but the rationale for the criteria should be provided.
- You mentioned that you checked extreme and erroneous values on lines 162 and 165, but you should show the exclusion criteria and their rationale.
- If ST or RST is only for one day around 169 lines, it is used for analysis, and if it is not more than two days, it is excluded from analysis, but the reason and rationale should be shown.
- I feel that "Sedentary behavior reduction programs ------- 120 (or 90) minutes for sons" from lines 341 to 344 may be difficult to derive from the results of this cross-sectional study. Should be modified or deleted.
Author Response
(Reviewer 3): This is an interesting study comparing sitting behavior and recreational screen time for Czech parents and children. Since it provides valuable knowledge, I believe that the possibility of publication will increase by making the following modifications.
- The conclusions of the abstract should be significantly revised. The reason is that it is a cross-sectional study, and since it is not based on causality, it is highly likely that the reader will misunderstand it.
Thank you for seeing the potential of our work. Thank you for pointing this issue out. We agree with your proposal. Consistent with similar comments by Reviewer 1 and 4, we have reformulated and clarified the text of the entire abstract and mitigated the wording that inferred causal relationships.
Abstract: Background: Excessive recreational screen time (RST) has been associated with negative health consequences already being apparent in preschoolers. Therefore, the aim of this study is to reveal parent-child sedentary behavior and RST patterns and associations with respect to the gender and age category of children and days of the week. Methods: Our cross-sectional three-cohort survey included 989 parent-child dyads with proxy-reported RST data collected during a regular school week during the spring and fall between 2013 and 2019. The parent-child RST (age and RST) relationship was quantified using Pearson's rP correlation coefficient. Results: Weekends are characterized by longer RST for all family members (daughters/sons: +34/+33 minutes/day, mothers/fathers: +43/+14 minutes/day) and closer parent-child RST associations than on weekdays. The increasing age of children is positively associated with an increase in RST on weekdays (+6.4/+7.2 minutes per year of age of the daughter/son) and weekends (+5.8/+7.5 minutes per year of age of the daughter/son). Conclusions: Weekends provide a suitable target for implementation of programs aimed at reducing excessive RST involving preferably all family members, which holds true both for families with preschoolers and for families with schoolchildren.
- The terms randomly are used on lines 88 and 90, but details should be given to what population they were randomly sampled from and how they were randomly sampled. Being a representative of the Czech people should show the rationale.
Thank you for pointing this issue out. This was our inaccuracy. Following this recommendation, we have added part “from the Czech Republic” and we have added an explanatory sentence.
Original: Invitation letters were sent to 3,540 families, of which 65.3% agreed to take part in the research (written informed consent received). Families were selected through two-stage random sampling. In the first stage, nine out of 14 administrative regions, three of each in the highest, middle, and lowest terciles for gross domestic product in Czechia, were randomly recruited. In the second stage, three public kindergartens located in rural areas and seven in urban locations and 15 public ‘basic schools’ located in rural areas and 36 public ‘basic schools’ in urban locations were randomly selected. The ‘basic schools’ in Czechia provide compulsory education from grades 1 to 9. The participants were largely white Caucasian (> 98%), which is representative of the ethnic demographics of Czechia [16].
Changed: Invitation letters were sent to 3,540 families from the Czech Republic, of which 65.3% agreed to take part in the research (written informed consent received). Families were selected through two-stage random sampling. In the first stage, nine out of 14 administrative regions, three of each in the highest, middle, and lowest terciles for gross domestic product in the Czech Republic, were randomly recruited. In the second stage, three public kindergartens located in rural areas and seven in urban locations and 15 public ‘basic schools’ located in rural areas and 36 public ‘basic schools’ in urban locations were randomly selected. Random selection of administrative regions and subsequently kindergartens and basic schools was performed using a random number generator from tables in computer memory. The ‘basic schools’ in the Czech Republic provide compulsory education from grades 1 to 9. The participants were largely white Caucasian (> 98%), which is representative of the ethnic demographics of the Czech Republic [22].
- Insufficient data on sitting behavior in line 99 states less than 4 days on weekdays and less than 1 day on weekends, but the rationale for the criteria should be provided.
- You mentioned that you checked extreme and erroneous values on lines 162 and 165, but you should show the exclusion criteria and their rationale.
- If ST or RST is only for one day around 169 lines, it is used for analysis, and if it is not more than two days, it is excluded from analysis, but the reason and rationale should be shown.
Thank you very much for these comments. You are right. We completely agree with you. To reflect on the comments, we have added the justification for the application of these criteria.
Added text: The criteria for inclusion of the data in the final set were as follows: (i) complete anthropometric data (year and month of birth, gender, body height and weight); (ii) attendance in kindergarten or basic school according to schedule (children), paid employment (parents other than maternity leave) on at least four school/working days a week; (iii) recording of the structure of sedentary behaviour (type, duration) on at least four school/working days and one weekend day; (iv) at least one parent-child pair (mother-child, father-child) per family. Inclusion criteria for the minimum number of days per week for a valid capture of sedentary behavior were determined according to the recommendations of thematically related studies in children [27,28] and adults [29,30]. … To maintain the comparability of family-related sedentary behavior data with previous studies focused on pedometer-based family physical activity [19,24,31], the same procedure for controlling and calculating anthropometric indicators and sedentary time/RST variables was retained.
- I feel that "Sedentary behavior reduction programs ------- 120 (or 90) minutes for sons" from lines 341 to 344 may be difficult to derive from the results of this cross-sectional study. Should be modified or deleted.
Thank you very much for this comment. We agree with your proposal and we have modified the mentioned text as follows.
Original: Sedentary behavior reduction programs should be implemented from early childhood, as the increase in sedentary behavior (or RST) between four and 16 years of age of offspring averages more than 100 (or 70) minutes for daughters and 120 (or 90) minutes for sons.
Changed: A significant increase in daily RST on weekends compared to school days for daughters and sons of all ages, which has replaced sedentary time on school days for schoolchildren, is a critical indicator of a sedentary lifestyle. Thus, weekends provide a suitable target for implementation of programs aimed at reducing excessive RST involving preferably all family members both in families with preschoolers and in families with schoolchildren.

Reviewer 4 Report
The authors dealt with an interesting research topic. To this end, they assessed a larger sample of three cohorts of 989 child-parent dyads.
Abstract: the first sentence is four lines long; please split and simplify.
Please define “recreational”; this is particularly important; do not expect that a reader can guess, what you mean with “recreational”; is “recreational” “good” or “bad”?
“…. and significant (rP = 0.167–0.395; p < 0.01) 17 parent-child RST associations.”; this part is difficult to understand.
“.. is significantly (p < 0.001)…”; please either report all statistical indices or no indices at all, but mere p-values are meaningless.
“
Conclusions: The replacement of 20 the lowest daily RST in members of families with preschoolers by a significant increase in daily RST for families with older children highlights the need for interventions targeting both parents and children…”.; in my opinion, this conclusion comes at a surprise and does not mirror what has been mentioned before in the Result section. Please revise.
Introduction: it appears that the very first paragraph is based on only 3 references; how come?
“…. Central Europe was not represented at all”; ok, but why should this be an issue?
Next, I strongly suggest to avoiding abbreviations; please do not overestimate the reader’s working memory capacity.
“However, …. of the week”. In my opinion, the authors present a broad variety of topics, but in a scattered and superficial fashion. I suggest to reducing the number of topic, and to describe in much more details the core topics. Further, as already mentioned for the Abstract, I strongly suggest to shortening the sentences such to keep the flow of reading and understanding smooth.
End of the paragraph; please formulate hypotheses: what kind of pattern of results do you expect and why? How might the results of the present study add to the current literature in an important fashion? Why should it be important to know, if a weekday-weekend-day is observable?
Method section. Sometimes, it is a matter of taste and judgement, though, I strongly recommend to introducing first the study design and study procedure. This holds particularly true as in my opinion the Participant section provides information, which should belong to the Result section.
Measures: Please organize this paragraph in a way that grasping the information gets easier. As it is now, it appears very difficult to grasp the key messages. Perhaps a table is very helpful with one column describing the key dimensions and another column describing the measures.
Results: Figures are particularly helpful! However, this is not the case for Figure 5; this looks particularly difficult and complex.
Further, it was difficult to understand why so many different statistical procedures were performed. To illustrate, the authors report results separately for gender, though, the Introduction section does not present and explain why this should be of theoretical or practical importance. Given this, also the whole Discussion section needs a thorough revision.
Author Response
(Reviewer 4): The authors dealt with an interesting research topic. To this end, they assessed a larger sample of three cohorts of 989 child-parent dyads.
Abstract: the first sentence is four lines long; please split and simplify.
Please define “recreational”; this is particularly important; do not expect that a reader can guess, what you mean with “recreational”; is “recreational” “good” or “bad”?
“…. and significant (rP = 0.167–0.395; p < 0.01) 17 parent-child RST associations.”; this part is difficult to understand.
“.. is significantly (p < 0.001)…”; please either report all statistical indices or no indices at all, but mere p-values are meaningless.
Conclusions: The replacement of 20 the lowest daily RST in members of families with preschoolers by a significant increase in daily RST for families with older children highlights the need for interventions targeting both parents and children…”.; in my opinion, this conclusion comes at a surprise and does not mirror what has been mentioned before in the Result section. Please revise.
Thank you for seeing the potential of our work and for your comments regarding the Abstract. We agree with your proposal. Consistent with similar comments by Reviewer 1 and 3, we have reformulated and clarified the text of the entire Abstract and mitigated the wording implying causality.
Abstract: Background: Excessive recreational screen time (RST) has been associated with negative health consequences already being apparent in preschoolers. Therefore, the aim of this study is to reveal parent-child sedentary behavior and RST patterns and associations with respect to the gender and age category of children and days of the week. Methods: Our cross-sectional three-cohort survey included 989 parent-child dyads with proxy-reported RST data collected during a regular school week during the spring and fall between 2013 and 2019. The parent-child RST (age and RST) relationship was quantified using Pearson's rP correlation coefficient. Results: Weekends are characterized by longer RST for all family members (daughters/sons: +34/+33 minutes/day, mothers/fathers: +43/+14 minutes/day) and closer parent-child RST associations than on weekdays. The increasing age of children is positively associated with an increase in RST on weekdays (+6.4/+7.2 minutes per year of age of the daughter/son) and weekends (+5.8/+7.5 minutes per year of age of the daughter/son). Conclusions: Weekends provide a suitable target for implementation of programs aimed at reducing excessive RST involving preferably all family members, which holds true both for families with preschoolers and for families with schoolchildren.
Introduction: it appears that the very first paragraph is based on only 3 references; how come?
Thank you very much for this comment. In line with a similar comment by Reviewer 1, we have reformulated sentences and added appropriate references so that the first paragraph is more strongly justified. (A smaller number of references occurred by reducing the references during our final conversion of the manuscript from the text editor to the IJERPH publication template.) In the findings, we have also highlighted the differences regarding the gender of the children to justify that we conducted statistical tests separately for boys (sons) and girls (daughters).
Original: Sedentary and screen time (ST), as one of the components of repetitive 24-hour behavior, such as physical activity, eating, and sleep, are closely linked to the health of children and young people [1]. A longer duration of ST (including television (TV) viewing, video games, and computer use) is significantly associated with higher cardiometabolic risks, lower fitness and self-esteem, and unfavorable body composition in children and young people; in other words, less sedentary behavior, in particular ST sedentary behavior, is better for optimal health [1]. Although studies in preschoolers are less frequent than in school-age children and adolescents, it is documented that increased TV viewing is associated with unfavorable measures of adiposity and decreased scores of psychosocial health and cognitive development [2]. An updated systematic review of the associations between sedentary behavior and health indicators in preschoolers [3] confirms the results of a previous systematic review [2], and continues to support the minimization of ST for disease prevention and health promotion in the early years of childhood, but also highlights the potential cognitive benefits of interactive non-ST sedentary behaviors such as reading and storytelling [3].
Changed: Sedentary and screen time, as one of the components of repetitive 24-hour behavior, such as physical activity, eating, and sleep, are closely linked to the health of children and young people [1–4]. A longer duration of screen time (including television viewing, video games, and computer use) is significantly associated with higher cardiometabolic risks, lower fitness, self-esteem, and unfavorable body composition and adiposity in preschoolers and schoolchildren [1–3] with different degrees of influence in girls and boys [4,5]. Although studies in preschoolers are less frequent than in schoolchildren and adolescents, it is documented that excessive screen time is associated with higher risk for overweight and obesity [6] and decreased scores of psychosocial health and cognitive development [2]. An updated systematic review of the associations between sedentary behavior and health indicators in preschoolers [3] confirms the results of a previous systematic review [2], and continues to support the minimization of screen time for disease prevention. Moreover, this review encourages the discovery of moderators of health promotion with respect to gender and highlights the potential cognitive benefits of interactive non-screen time behaviors such as reading and storytelling [3].
“…. Central Europe was not represented at all”; ok, but why should this be an issue?
Thank you for this comment and your close attention. This is an unfinished sentence. I would like to humbly apologize for this error that occurred during the conversion of the manuscript into the IJERPH publication template. We have reformulated and completed the mentioned sentence.
Original: However, while at least studies from Europe were represented in these systematic reviews, Central Europe was not represented at all.
Changed: It should be noted that only studies from Northern and Western European countries were represented in these systematic reviews [2,3]. Central and Eastern European countries were not included at all, although the prevalence of overweight and obesity as well as the screen time of preschoolers from Central and Eastern Europe is higher than among preschoolers from Northern and Eastern European countries [7,8].
Next, I strongly suggest to avoiding abbreviations; please do not overestimate the reader’s working memory capacity.
Thank you for pointing this issue out. You are right! Consistent with this comment, we have limited ourselves to using only the two abbreviations RST (recreational screen time) and BMI (Body Mass Index) throughout the manuscript and previous abbreviations (PC, TV, ST, DVDs) were written in full. The remaining abbreviations used are related to statistical analyses.
“However, …. of the week”. In my opinion, the authors present a broad variety of topics, but in a scattered and superficial fashion. I suggest to reducing the number of topic, and to describe in much more details the core topics. Further, as already mentioned for the Abstract, I strongly suggest to shortening the sentences such to keep the flow of reading and understanding smooth.
Many thanks for this comment. In line with similar comment by Reviewer 1, we have reformulated and clarified the text of the second and third paragraphs (including the less frequent use of the term however and pointing out possible differences with regard to the day of the week and the sex of preschoolers).
Original: However, preschoolers are a specific subgroup of children who are most dependent on their parents (guardians) and kindergarten teachers when 24-hour behavior is being organized. Therefore, parents tend to be labeled as “gatekeepers” of children's health-related behavior by determining what activities children do, as well as what resources and access they have at their disposal [4]. Of the 24-hour movement guidelines [5], it turns out that parents can best support compliance with sleep rules and ST restriction guidelines [6]. Subsequently, only compliance with the ST recommendations of the 24-hour movement guidelines was significantly associated with reduced odds of a high Body Mass Index (BMI) z-score, excess fat mass%, and visceral adipose tissue [7]. However, these findings [5,6] were revealed in children aged 5-17 [5,6] and 14-18-year-old adolescents [7]. Such findings are rare in preschoolers. However, ST appears to play a significant role in the prevalence of overweight in school-aged children and adolescents. Similar but even closer associations can be expected in preschoolers. First, however, it is necessary to analyze in detail the parent-child sedentary/ST and BMI of the association with respect to the gender of family members and days of the week.
Changed: Preschoolers are a specific subgroup of children who are most dependent on their parents (guardians) and kindergarten teachers when 24-hour behavior is being organized. Therefore, parents tend to be labeled as “gatekeepers” of children's health-related behavior by determining what activities children do, as well as what resources and access they have at their disposal [9]. Of the 24-hour movement guidelines [10], it turns out that parents can best support compliance with sleep rules and RST restriction guidelines [11]. Subsequently, only compliance with the RST recommendations of the 24-hour movement guidelines was significantly associated with reduced odds of a high Body Mass Index (BMI) z-score, excess fat mass%, and visceral adipose tissue [12]. These findings were revealed in children aged 5-17 [5,6] and 14-18-year-old adolescents [12], and it can be reasonably assumed that RST plays a significant role in the prevalence of overweight in schoolchildren and adolescents too. Similar but even closer associations can be expected in preschoolers. First, it is necessary to analyze in detail the parent-child sedentary behavior and RST patterns with respect to the gender of family members and days of the week. There exists evidence of variation in parent-child sedentary time according to gender of children and days of the week [13].
End of the paragraph; please formulate hypotheses: what kind of pattern of results do you expect and why? How might the results of the present study add to the current literature in an important fashion? Why should it be important to know, if a weekday-weekend-day is observable?
Thank you very much for these comments. We agree with your proposal and we have inserted our hypotheses immediately next to the formulated aims. We have also simplified the aims of the study and added a rationale for the hypotheses.
Original: In the countries of Central Europe, including Czechia, there are incomplete findings about sedentary behavior and especially the recreational ST of families with preschool children regarding the days of the week or the gender of family members. Therefore, the aim of this study is to reveal parent-child sedentary behavior and weekday-weekend recreational screen time (RST) patterns in family members with preschoolers in comparison with families with schoolchildren and adolescents (i.e., to compare the daily duration of total sedentary behavior/RST on weekdays and weekends of family members with preschoolers and families with older children, to find out the associations between RST/BMI/age of family members, and to quantify the change in RST with increasing age of children).
Changed and Added text: In the countries of Central Europe, including the Czech Republic, there are incomplete findings about sedentary behavior and especially the RST of families with preschool children regarding the days of the week or the gender of family members. Therefore, the aims of this study are to: (i) reveal parent-child sedentary behavior and weekday-weekend RST patterns in family members with preschoolers in comparison with families with schoolchildren and adolescents; (ii) find out the associations between RST/BMI/age of family members; (iii) quantify the change in RST with increasing age of children. In this study, we will test the hypotheses: (i) whether there is a difference in the daily sedentary time ((ii) in the daily RST) in daughters and sons (mothers and fathers) between weekdays and weekends in families separated by age categories of offspring; (iii) whether the parent-child association in daily RST varies with respect to the days of the week and gender; (iv) whether there is a significant association between daily RST and BMI ((v) daily RST and age) of children with respect to the days of the week. The sedentary behavior of children is influenced by many more variables besides parental behaviour on weekdays (e.g., kindergarten programme, school regimen, or participation in leisure-time organizations, clubs, sports), while at weekends their parents apparently might have more chances to stimulate health-enhancing behaviours in their offspring. Therefore, we expect closer association between the RST of parents and their children on weekends than on weekdays.
Method section. Sometimes, it is a matter of taste and judgement, though, I strongly recommend to introducing first the study design and study procedure. This holds particularly true as in my opinion the Participant section provides information, which should belong to the Result section.
Thank you for pointing this issue out. We agree with your proposal and we have changed the design of the Materials and Methods section. We have first introduced the section 2.1. Design of the Study and Ethnics, then 2.2. Selection of participants and Procedure. Finally, section 2.3. Measures precedes the 2.4 section on Statistical Analysis. We have moved the summary sample anthropometrics characteristics of the final data set of participants to the Result section. Please see the reworked Methods (as well as Results) section in the revised version of the manuscript.
Measures: Please organize this paragraph in a way that grasping the information gets easier. As it is now, it appears very difficult to grasp the key messages. Perhaps a table is very helpful with one column describing the key dimensions and another column describing the measures.
Thank you very much for this comment. To reflect on the comments, the text of the Measures section has been simplified and a table describing the items has been added. Please see Table 1 in the reworked Methods section in the revised version of the manuscript.
Added: Table 1. Parent-reported anthropometric measures and sedentary behavior-related items.
Results: Figures are particularly helpful! However, this is not the case for Figure 5; this looks particularly difficult and complex.
Thank you for this comment and your close attention. In reply to the comment, we have simplified Figure 5 and omitted redundant values.
Further, it was difficult to understand why so many different statistical procedures were performed. To illustrate, the authors report results separately for gender, though, the Introduction section does not present and explain why this should be of theoretical or practical importance. Given this, also the whole Discussion section needs a thorough revision.
Thank you for pointing this issue out. Following this recommendation, we have revised the text of the Discussion, added another limitation of our study, and reformulated the study's conclusions. Please see the Discussion section in the revised version of the manuscript (page 10, lines 319-322; page 10, lines 337-339; page 11, lines 367-370).

Round 2
Reviewer 3 Report
In this paper, the part I pointed out has been corrected almost perfectly. I felt that the other corrections were also essential corrections to help the reader's understanding. Therefore, we judge that this paper can be accepted.
Author Response
Reviewer Report 3:
(Reviewer 3): In this paper, the part I pointed out has been corrected almost perfectly. I felt that the other corrections were also essential corrections to help the reader's understanding. Therefore, we judge that this paper can be accepted.
Thank you very much.
Reviewer 4 Report
Overall, the quality of the manuscript improved; the authors took their job serious, and submitted a substantively modified manuscript.
In my opinion, the following points remain open.
The term “recreational” is problematic in the following ways. First, it remains unclear, if the authors use “recreational” as a synonym to “sedentary”. Second, following the Merriam-Webster dictionary (https://www.merriam-webster.com/dictionary/recreational ; retrieved April 16, 2021), “recreational” is defined as “done for enjoyment”, “used for pleasure”; typical synonyms are “amusing, delightful, diverting, enjoyable, and similar. Given this, if the authors really intend to say amusing, diverting, or enjoyable, then the whole story does not work. As such, I do kindly ask the authors to thoroughly revise their text. To illustrate: The original sentence is: “Excessive recreational screen time (RST) has been associated with negative health consequences already being apparent in preschoolers.”; the identical sentence is: “Excessive ENJOYABLE/AMUSING/DELIGHTFUL screen time (RST) has been associated with negative health consequences already being apparent in preschoolers.”; in my opinion, the second sentence is against the logic and plausibility.
Next, as a general rule, and following the guidelines of the American Psychological Association, the whole text should be put in past tense; this makes also sense, because the study has been accomplished; accordingly, sentences should be: “Weekends were characterized… “
“Weekends provide a suitable target for implementation of programs aimed at reducing excessive RST involving preferably all family members, which holds true both for families with preschoolers and for families with schoolchildren.”; this sentence is too long and grammatically problematic: “… family members, which….” is not correct.
Introduction: “… and unfavorable body composition and adiposity in preschoolers and schoolchildren [1–3_] with different degrees of influence in girls and 31 boys [4,5].”… what is the difference between “unfavorable body composition” and “adiposity”? What do you mean with “…with different degrees of influence in girls and boys…”; what are such “different degrees of influence”?
To my understanding, the Introduction section focused on research among preschoolers, though, the authors assessed a broad age-range from preschool-age to adolescence, as stated in the title; if true, the Introduction section must be completely revised.
“….while at weekends their parents apparently might have more chances to stimulate health-enhancing behaviours in their offspring. Therefore, we expect closer association between the RST of parents and their children on weekends than on weekdays…”; these claims appear somehow plausible, though, such statements need to be proofed with previous studies.
Overall, while I do acknowledge the efforts of the authors to improve the quality of the manuscript, in my opinion, substantive concerns as regards the underlying theoretical background, the focus and the aims of the study and style could not be satisfactorily adjusted.
Author Response
Reviewer Report 4:
(Reviewer 4): Overall, the quality of the manuscript improved; the authors took their job serious, and submitted a substantively modified manuscript. In my opinion, the following points remain open.
The term “recreational” is problematic in the following ways. First, it remains unclear, if the authors use “recreational” as a synonym to “sedentary”. Second, following the Merriam-Webster dictionary (https://www.merriam-webster.com/dictionary/recreational ; retrieved April 16, 2021), “recreational” is defined as “done for enjoyment”, “used for pleasure”; typical synonyms are “amusing, delightful, diverting, enjoyable, and similar. Given this, if the authors really intend to say amusing, diverting, or enjoyable, then the whole story does not work. As such, I do kindly ask the authors to thoroughly revise their text. To illustrate: The original sentence is: “Excessive recreational screen time (RST) has been associated with negative health consequences already being apparent in preschoolers.”; the identical sentence is: “Excessive ENJOYABLE/AMUSING/DELIGHTFUL screen time (RST) has been associated with negative health consequences already being apparent in preschoolers.”; in my opinion, the second sentence is against the logic and plausibility.
Next, as a general rule, and following the guidelines of the American Psychological Association, the whole text should be put in past tense; this makes also sense, because the study has been accomplished; accordingly, sentences should be: “Weekends were characterized… “
“Weekends provide a suitable target for implementation of programs aimed at reducing excessive RST involving preferably all family members, which holds true both for families with preschoolers and for families with schoolchildren.”; this sentence is too long and grammatically problematic: “… family members, which….” is not correct.
Thank you for seeing the potential of our work and for your comments regarding the Abstract.
Consistent with these comments, we have reformulated and clarified the text of the entire abstract. The entire text of the abstract was rephrased in the past tense. The last sentence of the abstract has been shortened and reformulated. In line with a comment by Editor, we have omitted the term “three-cohort”. However, we have decided to keep the term “recreational screen time”, as it is a “terminus technicus” in this field of research and used, e.g. in the 2020 WHO PA guidelines (doi:10.1136/bjsports-2020-102955)
Changed: Abstract: Background: Excessive recreational screen time (RST) has been associated with negative health consequences already being apparent in preschoolers. Therefore, the aim of this study was to reveal parent-child sedentary behavior and RST patterns and associations with respect to the gender and age category of children and days of the week. Methods: Our cross-sectional survey included 1175 parent-child dyads with proxy-reported RST data collected during a regular school week during the spring and fall between 2013 and 2019. The parent-child RST (age and RST) relationship was quantified using Pearson's rP correlation coefficient. Results: Weekends were characterized by longer RST for all family members (daughters/sons: +34/+33 minutes/day, mothers/fathers: +43/+14 minutes/day) and closer parent-child RST associations than on weekdays. The increasing age of children was positively associated with an increase in RST on weekdays (+6.4/+7.2 minutes per year of age of the daughter/son) and weekends (+5.8/+7.5 minutes per year of age of the daughter/son). Conclusions: Weekends provide a suitable target for implementation of programs aimed at reducing excessive RST involving not only children, but preferably parent-child dyads.
Introduction: “… and unfavorable body composition and adiposity in preschoolers and schoolchildren [1–3_] with different degrees of influence in girls and 31 boys [4,5].”… what is the difference between “unfavorable body composition” and “adiposity”? What do you mean with “…with different degrees of influence in girls and boys…”; what are such “different degrees of influence”?
Thank you for pointing this issue out. Following this recommendation, we have we omitted the redundant term "unfavorable body composition" and replaced the part "different degrees of influence in" with "with obvious differences between girls and boys."
To my understanding, the Introduction section focused on research among preschoolers, though, the authors assessed a broad age-range from preschool-age to adolescence, as stated in the title; if true, the Introduction section must be completely revised.
Thank you very much for this comment and pointing this issue out. In line with a similar comment by Editor, we have revised the text of the Introduction section. We have kept the first paragraph and amended paragraphs two and three, in which we devoted more space to schoolchildren and adolescents compared to the previous version. Please see the reworked section in the revised version of the manuscript (and comments to the Editor below).
“….while at weekends their parents apparently might have more chances to stimulate health-enhancing behaviours in their offspring. Therefore, we expect closer association between the RST of parents and their children on weekends than on weekdays…”; these claims appear somehow plausible, though, such statements need to be proofed with previous studies.
Many thanks for this comment. In reply to the comment, we have added appropriate references to the previous two studies [21,25].
- Sigmundová, D.; Badura, P.; Sigmund, E.; Bucksch, J. Weekday-weekend variations in mother-/father–child physical activity and screen time relationship: a cross-sectional study in a random sample of Czech families with 5- to 12-year-old children. J. Sport Sci. 2018, 18, 1158–1167. DOI: 10.1080/17461391.2018.1474951
- Sigmundová, D.; Sigmund, E.; Vokáčová, J.; Kopčáková, J. Parent-Child Associations in Pedometer-Determined Physical Activity and Sedentary Behaviour on Weekdays and Weekends in Random Samples of Families in the Czech Republic. J. Environ. Res. Public Health2014, 11, 7163-7181. https://doi.org/10.3390/ijerph110707163